# Lack of abundant core virome in *Culex* mosquitoes from a temperate climate region despite a mosquito species-specific virome

Lander De Coninck,[1] Alina Soto,[2] Lanjiao Wang,[2] Katrien De Wolf,[3,4] Nathalie Smitz,[5] Isra Deblauwe,[3] Karelle Celes Mbigha Donfack,[1] Ruth Müller,[3] Leen Delang,[2] Jelle Matthijnssens[1]

**ABSTRACT**    In arthropod-associated microbial communities, insect-specific viruses (ISVs) are prevalent yet understudied due to limited infectivity outside their natural hosts. However, ISVs might play a crucial role in regulating mosquito populations and influencing arthropod-borne virus transmission. Some studies have indicated a core virome in mosquitoes consisting of mostly ISVs. Employing single mosquito metagenomics, we comprehensively profiled the virome of native and invasive mosquito species in Belgium. This approach allowed for accurate host species determination, prevalence assessment of viruses and *Wolbachia*, and the identification of novel viruses. Contrary to our expectations, no abundant core virome was observed in *Culex* mosquitoes from Belgium. In that regard, we caution against rigidly defining mosquito core viromes and encourage nuanced interpretations of other studies. Nonetheless, our study identified 45 viruses of which 28 were novel, enriching our understanding of the mosquito virome and ISVs. We showed that the mosquito virome in this study is species-specific and less dependent on the location where mosquitoes from the same species reside. In addition, because *Wolbachia* has previously been observed to influence arbovirus transmission, we report the prevalence of *Wolbachia* in Belgian mosquitoes and the detection of several *Wolbachia* mobile genetic elements. The observed prevalence ranged from 83% to 92% in members from the *Culex pipiens* complex.

**IMPORTANCE**    *Culex pipiens* mosquitoes are important vectors for arboviruses like West Nile virus and Usutu virus. Virome studies on individual *Culex pipiens*, and on individual mosquitoes in general, have been lacking. To mitigate this, we sequenced the virome of 190 individual *Culex* and 8 individual *Aedes japonicus* mosquitoes. We report the lack of a core virome in these mosquitoes from Belgium and caution the interpretation of other studies in this light. The discovery of new viruses in this study will aid our comprehension of insect-specific viruses and the mosquito virome in general in relation to mosquito physiology and mosquito population dynamics.

**KEYWORDS**    virome, mosquito, *Culex pipiens*, single mosquito metagenomics, insect-specific viruses

Address correspondence to Jelle Matthijnssens, jelle.matthijnssens@kuleuven.be.

The authors declare no conflict of interest.

See the funding table on p. 18.

As a result of the democratization of next-generation sequencing (NGS) techniques, the number of metagenomic studies on mosquitoes has exponentially grown, and simultaneously, the discovery of new viruses has soared (1–3). However, only a small fraction of these new viruses is capable of infecting humans and mammals; these viruses are called arthropod-borne viruses or arboviruses. The majority of newly discovered mosquito-borne viruses are either insect-specific viruses (ISVs), replicating solely in the mosquito host, passerby viruses from the mosquito diet, or viruses infecting mosquito

parasites (1, 4). ISVs are considered as one of the most abundant components of arthropod-associated microbial communities but remain largely unstudied primarily due to their lack of infectivity outside their natural hosts. Despite this, they are believed to play an important role in regulating mosquito population dynamics and they have been shown to influence arbovirus transmission (5–8). Interestingly, multiple studies across different mosquito hosts have observed that the same ISVs are often present in many mosquitoes from the same species, which implies the existence of a set of widely distributed species-specific ISVs, often referred to as a "core virome" (9–11). Such a core virome is believed to have co-evolved with their hosts over an extended period of time, thereby having a profound impact on their biology. It might also modulate the ability of a host to serve as a competent vector for arboviruses.

Most mosquito research focuses on *Aedes* mosquitoes in tropical countries as this genus includes major vectors of arboviruses like dengue virus, yellow fever virus, and Zika virus. *Culex* mosquitoes are, however, also a common vector of human and animal diseases, including arboviruses such as West Nile virus and Usutu virus (12). Major efforts to sequence the virome of these genera have been made in the past few years (3). However, despite the potential advantages of single mosquito virome studies (e.g., more accurate virus prevalence determination in the mosquito population, linking viruses to bloodmeals of different hosts, and, furthermore, supplementing morphological mosquito species identification), there are only a handful of studies describing the virome of individual mosquitoes (9, 13, 14).

Furthermore, apart from ISVs, bacteria, and particularly *Wolbachia*, can also have an influence on (arbo)virus replication in mosquitoes (15, 16). Besides reducing (arbo)virus replication, *Wolbachia* has also been shown to reduce the fitness and reproduction capacity of mosquitoes due to cytoplasmic incompatibility. The genes responsible for this effect lie within prophage regions of the *Wolbachia* genome (17–19). Additionally, the World Mosquito Program (WMP) recently used *Wolbachia*-infected *Aedes aegypti* to showcase the potential of arbovirus control by commensal microbes (20, 21). The goal of the WMP is to eradicate mosquito-borne diseases like dengue fever, yellow fever, and Chikungunya in Latin America, Asia, and Oceania using experimentally infected *Ae. aegypti* mosquitoes (22). Interestingly, *Wolbachia* is naturally present in the *Culex pipiens* populations in Europe (23, 24).

In the present investigation, the individual viromes of 190 mosquitoes native to Belgium were sequenced, including specimens of both *Culex pipiens* biotypes (*Cx. p. pipiens* and *Cx. p. molestus*) and of *Culex torrentium* (25). *Culex pipiens* and *Cx. torrentium* are both common and widespread in human habitats and occur in sympatry in Belgium (26). Additionally, the viromes of eight invasive *Aedes japonicus japonicus* mosquitoes were characterized as comparison. Finally, we described the prevalence of *Wolbachia* and its mobile genetic elements in the Belgian *Culex* population.

## MATERIAL AND METHODS

### Mosquito collection

Mosquitoes were collected with BG-Sentinel, Mosquito Magnet, or the Frommer Updraft Gravid Traps across Belgium between 2019 and 2020 in the framework of the MEMO project [Monitoring of Exotic MOsquito species in Belgium (27); Institute of Tropical Medicine] and a collection program of the Mosquito Virology Team at KU Leuven. After collection, mosquitoes were stored dry at −80°C in single tubes until further processing. Molecular identification on the species and biotype level was achieved by regular (q)PCR techniques based on the cytochrome c oxidase I gene, the acetylcholinesterase 2 gene, and the CQ11 microsatellite region as described in Vanderheyden et al. (28) and Wang et al. (29). In total, 198 mosquitoes were collected for this study (Fig. 1A), of which none showed visible signs of a recent bloodmeal.

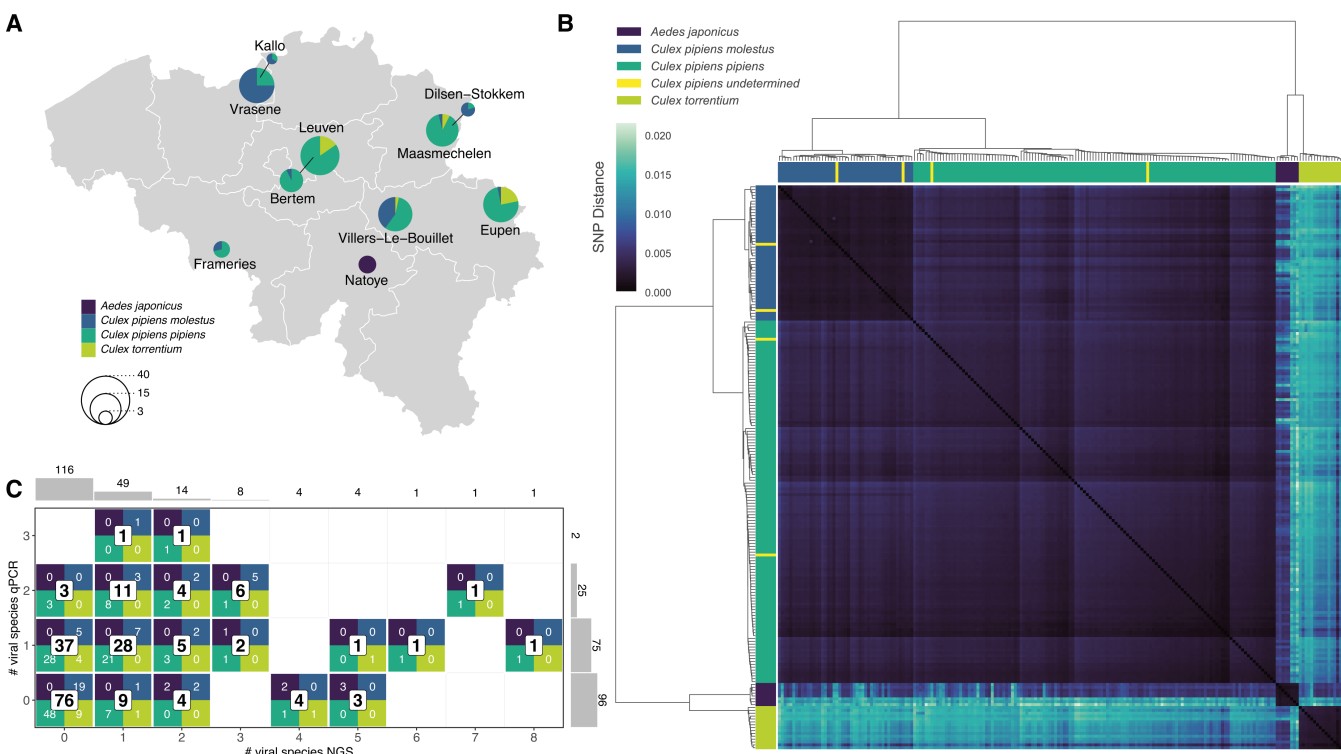

**FIG 1** Mosquito species and biotype determination and overview of collection sites. (A) Map of Belgium showing all collection sites and the number of captured mosquitoes per taxa at each site represented by a pie chart. (B) Pairwise single nucleotide polymorphism (SNP) comparison, clustered based on the Ward distance. Colored bars on top and on the left side show the (q)PCR species determination of each sample, while the heatmap displays the SNP distance between all samples. A SNP distance of 0.01 means that 1% of comparable sites had SNPs. (C) Graph showing the total amount of mosquitoes studied (additionally subdivided per species and biotype), broken down by the number of viruses that were detected in each mosquito with metagenomics (*x*-axis) and qPCR (*y*-axis). Two histograms on the outside show the total number of mosquitoes per virus count for each method.

## Sample processing and sequencing

The NetoVIR protocol, a standardized and reproducible protocol for viromics, was used to purify the samples for viral particles (30). In short, single mosquitoes were diluted in phosphate buffered saline (PBS) and homogenized in a Minilys Tissue Homogenizer (Bertin Instruments) with 2.8 mm zirconium oxide beads (Precellys) at 3,000 rpm for 1 min. For each processed batch of mosquitoes, a negative control consisting of only PBS was taken along. Next, the samples were centrifuged at 17,000 × *g* for 3 min, and 150 µL supernatant of each sample was subsequently filtered through a 0.8 µm filter (Sartorius). This filtrate was treated with a mix of Benzonase (50 U, Novagen) and Micrococcal nuclease (2,000 U, New England Biolabs) to digest remaining free-floating eukaryotic and bacterial nucleic acids. Viral RNA and DNA were then extracted using the QIAamp Viral RNA Mini Kit (QIAGEN) according to the manufacturer's instructions but without carrier RNA. DNA and RNA were amplified using the Complete Whole Transcriptome Amplification kit (WTA2, Merck), and resulting PCR products were further purified and prepared for sequencing with the Nextera XT kit (Illumina). The final sequencing libraries were cleaned up with Agencourt AMPure XP beads (Beckman Coulter, Inc.) using a 1:1 ratio. Finally, paired-end sequencing was performed on the Nextseq 550 platform (Illumina) for 300 cycles (2 × 150 bp) with an estimated average of 10 million reads per sample.

## Mosquito species identification by NGS data

Although our samples were enriched for virus-like particles, a significant percentage of the obtained reads were still of mosquito genome origin (see below). These reads

were used to validate and resolve inconclusive mosquito species identifications resulting from standard (q)PCRs. A split k-mer analysis was performed on the obtained non-viral NGS reads to determine the species of each sample onto the biotype level (13, 31). Pairwise single nucleotide polymorphism (SNP distances were calculated on the raw read files (which were dominated by mosquito reads); samples were subsequently hierarchically clustered based on the Ward distance and placed ultimately into four groups [the number of mosquito species/biotypes observed by (q)PCR identification] (Fig. 1B).

## Read processing and contig taxonomic assignment

Obtained raw reads were processed with the ViPER script, with the "triple assembly" setting enabled (32). Briefly, raw reads were trimmed with Trimmomatic v.0.39 (33) for WTA2 primers and Nextera XT adapters as well as low-quality bases. Trimmed reads were mapped to a set of complete mosquito genomes (*Aedes aegypti*: GCA_002204515.1, *Aedes albopictus*: GCA_006496715.1, *Culex quinquefasciatus*: GCA_015732765.1) with Bowtie2 (34) on the very sensitive setting to remove host reads, and the remaining reads were subsequently assembled into contigs using metaSPAdes v.3.15.2 (35). To remove redundancy in the data, contigs from all 198 samples and 10 controls were clustered together at 95% nucleotide identity over a coverage of 85% of the shortest sequence, using BLAST (36) and the clustering algorithm published with CheckV (37). Contigs were taxonomically assigned using Diamond v.2.0.9 (38) with the NCBI nr database (accessed 17 March 2023), KronaTools v.2.8 (39), and TaxonKit v.0.8.0 (40), employing a lowest common ancestor approach.

## Eukaryotic virome analysis

The trimmed reads of each individual sample were mapped back to the set of clustered, non-redundant contigs with bwa-mem2 (41). A contig was considered present in a sample if the contig was covered by the reads for at least 50% of its length as calculated by CoverM v.0.6.1 (42). The resulting read counts for each sample to each contig were stored in a matrix (abundance table) and used for further analyses in R. First, contaminating contigs were removed by the prevalence method of the decontam package (43). Virome diversity and richness analyses were further performed on contigs larger than 1,000 nucleotides with the phyloseq (44), vegan (45), and ComplexHeatmap (46) packages. To calculate alpha (diversity within a sample) and beta diversity (diversity between samples), the viral abundance matrix was rarified 1,000 times to a sequencing depth of 152 reads for each sample (the lowest number of reads at a natural break in the data that removes less than 5% of the samples), and the average value across diversity calculations from these 1,000 rarified abundance matrices was taken (47). Wilcoxon tests were used to compare alpha diversity means across the different species. We also applied a permutational multivariate analysis of variance (adonis2 from the vegan R package) on the Bray-Curtis dissimilarity matrix to test whether differences in virome abundances and composition are explained by mosquito species/biotype and/or location. To visualize virome communities based on mosquito species/biotype, principal coordinate (PCoA) and non-metric multidimensional scaling (NMDS) analyses were performed on the dissimilarity matrix.

## RT-qPCR of interesting viruses

A panel of highly prevalent and/or abundant (insect-specific) viruses (see above) was selected based on the NGS data to quantify these viruses in all samples with RT-qPCR. Specific primers and TaqMan probes (see Table S1) were designed with PriMux (48) in the RNA-dependent RNA polymerase (RdRP) region of the recovered near-complete genomes. The remaining extracts of the samples were diluted with RNase-free water to have sufficient volume for all RT-qPCRs before aliquoting the extracts to separate PCR plates to limit freeze-thaw cycles. For each RT-qPCR, the total reaction volume per sample was 20 µL, which consisted of 5 µL TaqMan Fast Virus 1-Step Master Mix (Thermo

Fisher), 2 µL forward and reverse primer (10 µM), 1 µL probe (5 µM), and 5 µL viral RNA extraction. To determine the viral genome copy number, each reaction was accompanied by a 10-fold dilution series of oligonucleotide standards with known concentration (from $10^3$ to $10^8$ copies). Genome copy number calculations were performed in Applied Biosystems' Design and Analysis v.2.6.0 software. The viral genome copy number for each sample was recalculated in accordance with the dilution factor of the sample to obtain a viral genome copy number per whole mosquito body.

## Virus phylogenetics

Open reading frames (ORFs) of near-complete viral genomes were predicted by NCBI's ORFfinder tool (https://www.ncbi.nlm.nih.gov/orffinder/). ORFs encoding for the RdRP protein were selected for phylogenetic analysis. These complete RdRP protein sequences were searched against the NCBI nr database with BLASTp, and for each distinct viral species in the BLAST result, one representative RdRP sequence was downloaded if the query coverage was higher than 70% (with exception of the *Endornaviridae* for which we selected a query coverage of at least 30%). In addition, we downloaded the RdRP sequences of representative viral species as classified by the International Committee on the Taxonomy of Viruses (ICTV) for each viral taxonomic group we encountered in our data set.

Duplicates in the sequence sets were removed with the BBMap tools suite (https://sourceforge.net/projects/bbmap/), before aligning them with MAFFT v.7.490 using the E-INS-I algorithm (49). Resulting alignments were automatically trimmed with Trimal v.1.4 (50) on the gappy-out setting. Maximum likelihood phylogenetic trees were subsequently constructed with IQ-TREE 2 (51), using automated model selection (models were restricted to models available in PhyML) and 1,000 ultrafast bootstraps. Phylogenetic trees were midpoint rooted and visualized in R with phytools and ggtree (52, 53).

## Co-occurrence analysis for viral segments

Based on the idea presented by Batson et al. that viral segments will co-occur in samples where the same segmented virus is present (13), we performed correlation tests on the RdRP segments of the identified orthomyxoviruses. In practice, the abundance table of this study (see above) was divided by the contig length and subsequently used to calculate the Spearman correlation coefficient between the RdRP segments of each identified orthomyxovirus and all other contigs. Afterward, the contigs with the highest correlation coefficient were manually curated by looking at both BLASTx results and contig coverage to identify the remaining unknown segments. The script for this co-occurrence analysis is available at https://github.com/LanderDC/co-occurrence.

## Phageome analysis and *Wolbachia* prevalence estimation

The set of contigs larger than 1,000 nucleotides (see above) was analyzed with Virsorter2 (54) and CheckV (37) to discover bacteriophage genomes and estimate their completeness, respectively. Contigs that were not predicted to be a eukaryotic virus by our earlier analysis and that were more than 20% complete as predicted by CheckV were regarded as reliable bacteriophage contigs.

To assess the prevalence of *Wolbachia* in the Belgian mosquito population, we mapped the set of trimmed reads to the *Wolbachia* strain *w*Pip genome (accession number AM999887.1). To consider *Wolbachia* present in a sample, the horizontal coverage of the mapped *Wolbachia* genome had to exceed 5%. The resulting BAM files were also used to evaluate the presence of either phage WO prophage sequences or real WO viral particles. The sequencing depth at each position of the *Wolbachia* genome was calculated with samtools (55), and subsequently, the average sequencing depths in prophage regions and non-prophage regions (excluding two rRNA genes) in the *Wolbachia* genome were computed. Next, the ratio of the average depth in prophage regions over the average depth in non-prophage regions indicated if true phage WO

particles were present in the sample or if it was merely the prophage regions that were sequenced (depth ratio cutoff >3).

## RESULTS

### Mosquito species identification

Mosquitoes were captured across 10 locations in Belgium between 2019 and 2020 (Fig. 1A). Firstly, it was imperative to correctly identify which species each mosquito sample belonged to, as this might influence their microbiome (9). Therefore, we characterized each mosquito with (q)PCR at the species and biotype level. However, we could not resolve the biotype of four *Culex pipiens* mosquitoes with these established methods. Therefore, we employed a pairwise comparison of SNP distances from a split k-mer analysis on our raw sequencing data which predominantly contained mosquito host sequences. Of note, a single sample (MEMO011; *Culex torrentium* defined by PCR) was removed from the split k-mer analysis because it had less than 5,000 sequencing reads that mapped to our set of mosquito genomes (see Material and Methods for accession numbers). After hierarchically clustering the samples based on the Ward distance, we could distinguish four groups corresponding with the molecular identification of the mosquitoes. This eventually revealed the biotype of the previously undetermined *Culex pipiens* mosquitoes (Fig. 1B). In total, we further analyzed the virome of 8 *Aedes japonicus* (an invasive established mosquito species in Belgium), 47 *Culex pipiens molestus*, 127 *Culex pipiens pipiens,* and 16 *Culex torrentium* mosquitoes.

### Eukaryotic virome analysis

After sequencing, we obtained 1,069,326,190 reads of which 273,050,395 remained after trimming and the removal of host sequences. This set of trimmed, nonhost reads was subsequently assembled into 133,323 contigs larger than 500 bp. These contigs were clustered on 95% nucleotide identity and 85% coverage of the shortest sequence to remove redundancy, resulting in a set of 62,957 non-redundant contigs.

In order to have high certainty about the detected viruses in our samples, we applied stringent criteria on our non-redundant contig set. For the eukaryotic virome analysis, we filtered out contigs annotated as viral by Diamond and KronaTools, and only considered contigs larger than 1,000 nucleotides. The remaining set of viral contigs was manually curated to remove possible endogenous viral elements and to resolve the annotation of different segments from divergent, segmented RNA viruses. Using these criteria, no viruses were detected in 116 mosquitoes. Nevertheless, 49 mosquitoes contained one virus, while 33 mosquitoes, including all *Aedes japonicus* mosquitoes, harbored multiple viral species. A single *Culex pipiens pipiens* sample had eight distinct viruses identified (Fig. 1C). Overall, this corresponded to 147 viral contigs with 12,533,786 viral reads across all samples.

The observed viral species belonged to 23 different viral families, of which the relative abundance in the different mosquito species is shown in Fig. 2A. Each mosquito species had a distinct set of viral families, with little overlap. On the other hand, the relative abundance of the detected viral families, based on the collection site, showed that some viral families were present at multiple locations, e.g., *Chrysoviridae, Nodaviridae,* and *Orthomyxoviridae* (Fig. 2B).

Additionally, we calculated alpha and beta diversity on the eukaryotic virome of the samples reported with at least one eukaryotic virus. Looking at alpha diversity (richness, Shannon, and Simpson indices) in the different mosquito species, it was clear that the *Aedes japonicus* had a higher viral diversity than both *Culex pipiens* biotypes (Fig. 3A). The higher average alpha diversity in the *Culex torrentium* samples can be explained by the presence of multiple viral species infecting fungi (see below) and a small number of *Culex torrentium* samples. For the beta diversity analysis, the Bray-Curtis dissimilarity was calculated based on the abundance of eukaryotic viral species in our data set. This dissimilarity metric was used in PCoA and NMDS ordination analyses (Fig. 3B and C).

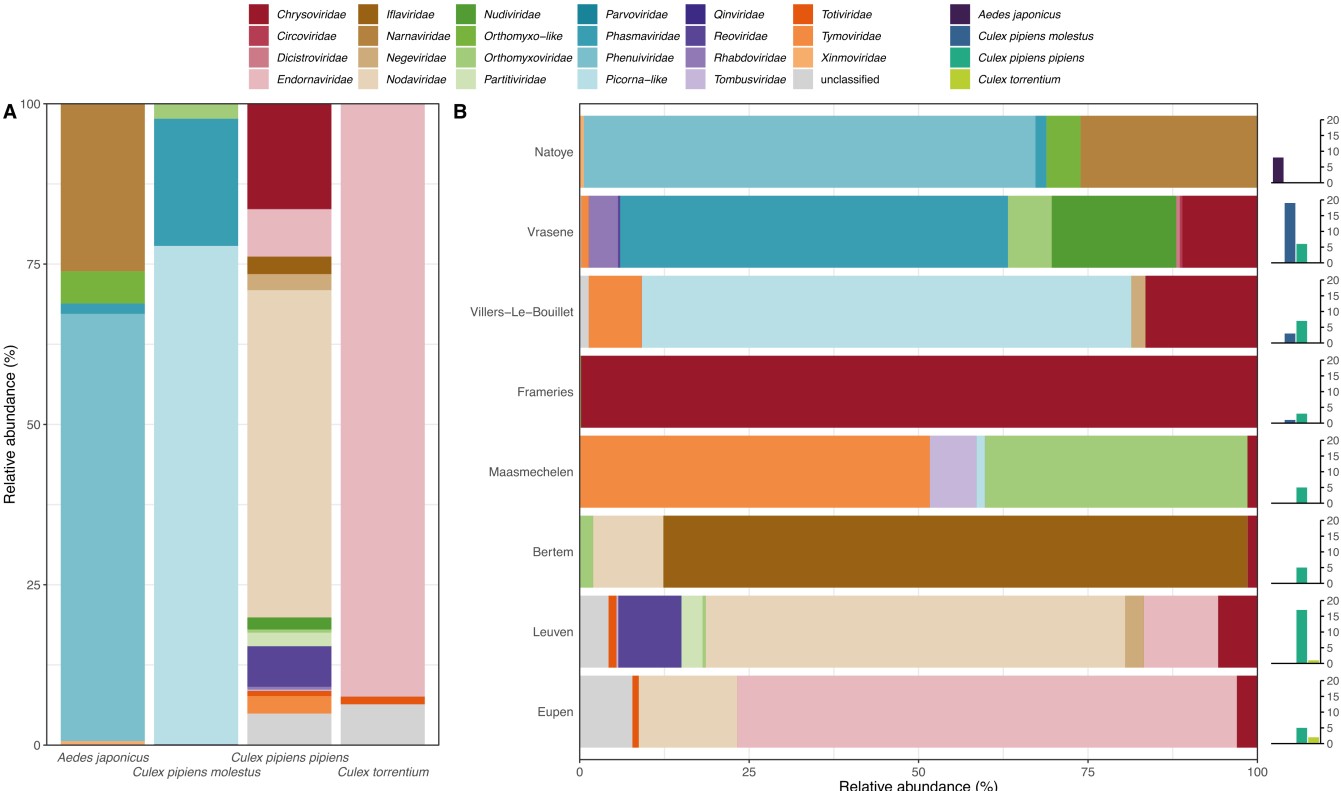

FIG 2 Relative abundance of viral families per mosquito species and location. The bar charts show the viral diversity at family level in Belgian mosquitoes (A) in the different collected mosquito species and (B) at the different collection locations. In panel B, the number of each captured mosquito species per location is added on the right.

Subsequently, an Adonis test showed that the mosquito species/biotype explained 25% of the variation in the virome between mosquitoes (*P*-value < 0.001). Furthermore, if we removed the singleton samples in the NMDS ordination of Fig. 3C, a clear separation between the mosquito genera appeared in the resulting NMDS plot (see Fig. S1).

In addition, we constructed a heatmap of our eukaryotic virome data set showing the abundance of each detected viral species in each sample (Fig. 4). We could detect 45 viruses, of which 42 were RNA viruses, 2 were single-stranded DNA (ssDNA) viruses, and 1 was a double-stranded DNA (dsDNA) virus belonging to the family *Nudiviridae*. In the heatmap, the viruses were alphabetically sorted on the Baltimore classification of their putative viral family (dsDNA, ssDNA, dsRNA, positive and negative ssRNA), while the samples were clustered based on their virome composition using the Bray-Curtis dissimilarity. This exhibits a clear clustering pattern according to mosquito species/ biotype and less according to the collection location, confirming the results from the Adonis test (see above). The average amino acid identity (AAI; shown on the left in Fig. 4), which was calculated with BLASTx across multiple viral segments or fragments of the same genome, indicates that we discovered 28 novel viruses out of the 45 detected viruses (based on a cutoff of 95% AAI).

For each virus, we attempted to infer their host species based on the isolation source of their best BLAST hit and their AAI. For example, if the detected virus was related to a virus sequenced in another insect species with low AAI, we assigned the mosquito to be the 'likely' host. If the closest BLAST hit was isolated from a mosquito species, we assigned the determined host species to be the "highly likely" mosquito. On the other hand, when the host of the closest BLAST hit was a fungus and we found fungal reads in our data co-occurring with the presence of that virus, we considered that virus to be likely a fungus-infecting virus. We could link 36 viruses to the mosquito as the probable host, six viruses with a fungus as the host, but for three viruses (a parvovirus, Keenan

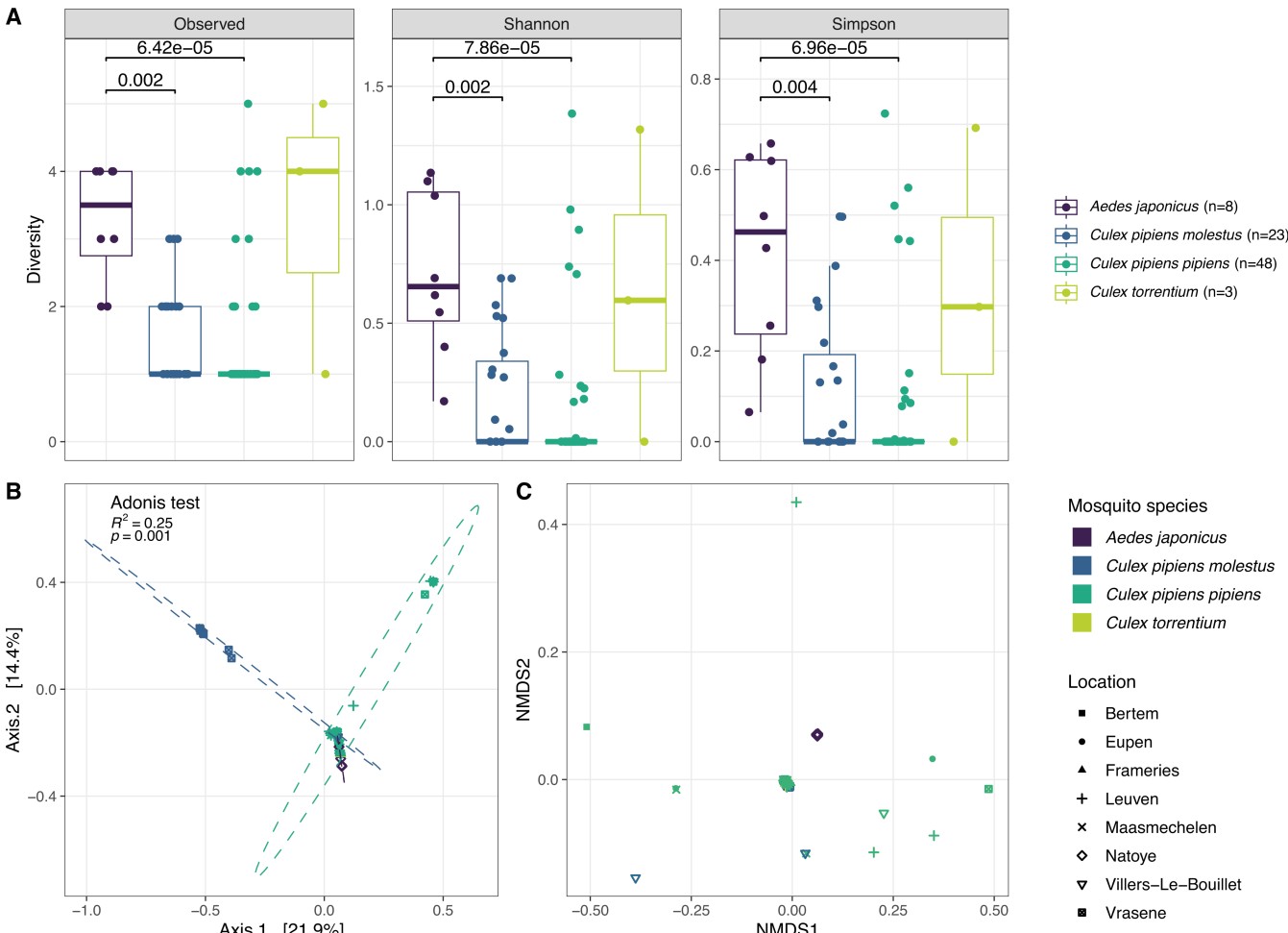

**FIG 3** Alpha and beta diversity of the eukaryotic virome in the Belgian mosquito samples. (A) The observed richness, Shannon, and Simpson alpha diversities were calculated based on the eukaryotic virome for samples with at least one virus present. A Wilcoxon test revealed the significant difference in viral diversity between the *Aedes japonicus* and *Culex pipiens* species. (B) Results of PCoA showing the first two components which together represent 36.3% of the variation in the data. Points are colored based on mosquito species/biotype and shaped based on their collection location (Adonis test on mosquito species: $P = 0.001$, $R^2 = 0.25$). (C) NMDS plot of the eukaryotic virome.

toti-like virus, and *Arthrocladiella mougeotii* alphaendornavirus), it was uncertain what their host species were. The closest hit to the parvovirus was not an insect virus, and for Keenan toti-like virus and the *Arthrocladiella mougeotii* alphaendornavirus (both expected to be fungal viruses), we could not find any fungal reads in the respective samples. In addition, we could not assemble a full genome for these three viruses, which complicates an accurate viral taxonomic assignment and determination of host species. Interestingly, we mostly found highly abundant fungal viruses in mosquitoes that also harbored reads belonging to entomopathogenic fungi (e.g., Microsporidia, Chytridiomycota). Meanwhile, we did not observe any fungal viruses in the samples with only Ascomycota or Basidiomycota reads, two fungal phyla which could be pathogenic as well as non-pathogenic for mosquitoes (56). The presence of these four fungal phyla was extrapolated from the taxonomic annotation of our contigs by Diamond.

## RT-qPCR analysis of potential core virome

As it is difficult to make quantitative claims based on NGS data from metagenomic sequencing, the six most abundant and/or prevalent (non-fungal) viruses in the *Culex* mosquitoes were selected for a quantitative analysis with qPCR (Fig. 4, shown in red). These viruses were Xanthi chryso-like virus (XCV), Daeseongdong virus 2 (DV2), Hubei

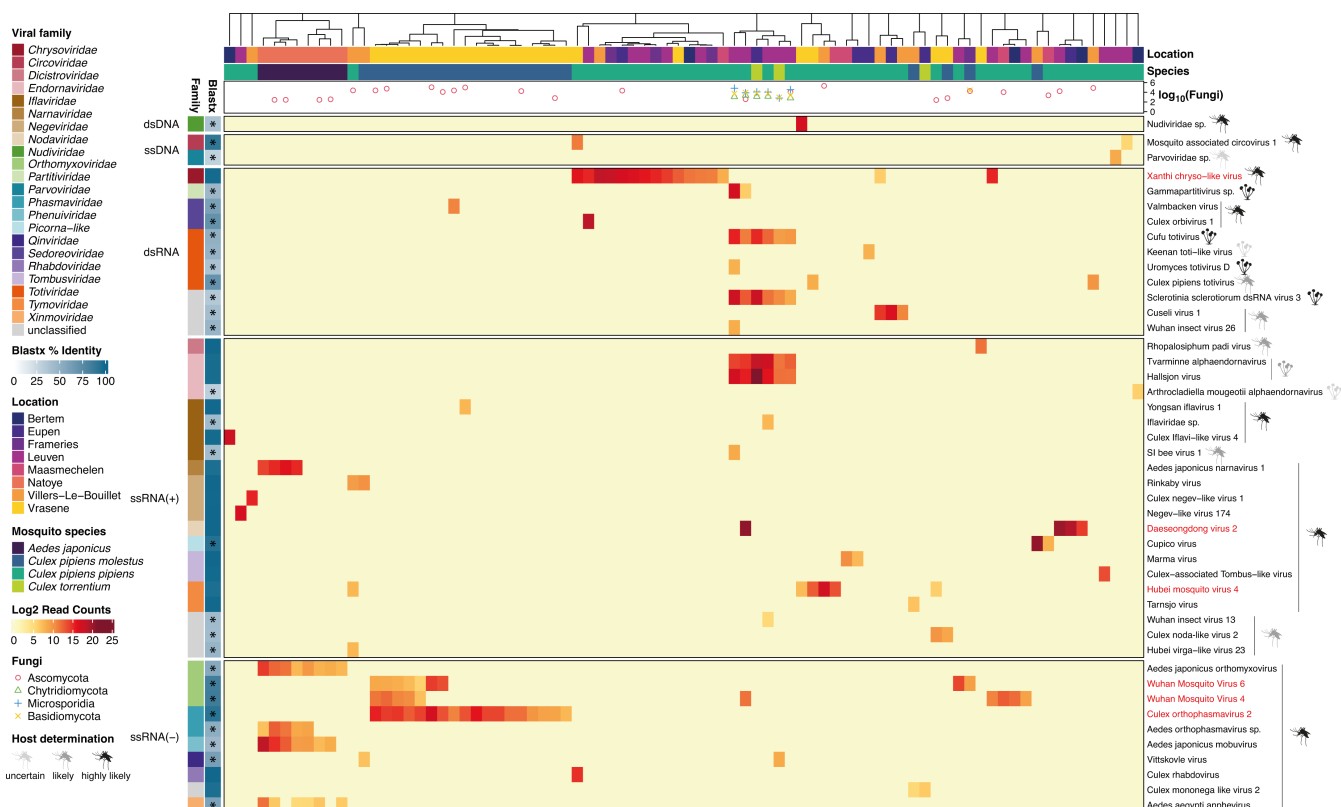

**FIG 4** Heatmap of individual viruses present in Belgian mosquitoes. The binary logarithm read count of each virus in each sample is shown. Viruses are alphabetically sorted on their assigned taxonomy on family level. They are further subdivided based on their genome organization. Average BLASTx percentage identities are shown on the left for each virus; novel viruses (<95% identity) are indicated with an asterisk. Samples are clustered based on the Bray-Curtis dissimilarity between their eukaryotic viromes. On top of the heatmap, the collection location and mosquito species of each sample are displayed, revealing a clustering pattern based mainly on mosquito species. The common logarithm of the read count for four different fungal phyla is shown for each sample. Furthermore, the host species (mosquito or fungus) and how likely this designation is are indicated next to each virus. The viruses colored in red were selected for quantitative analysis by qPCR based on their abundance and prevalence.

mosquito virus 4 (HMV4), *Culex orthophasmavirus* 2 (CPV), and Wuhan mosquito virus 4 and 6 (WMV4, WMV6). Hypothesizing that these viruses might be part of a core virome in *Culex* mosquitoes (9), we developed and performed a TaqMan RT-qPCR for these viruses on the remaining diluted nucleic acid extracts of the samples.

In Fig. 5, the viral genome copy numbers are shown per virus and per mosquito species. As generally assumed, the qPCR was more sensitive than the metagenomic sequencing, since more samples were positive for all selected viruses with qPCR compared to the NGS. Interestingly, for CPV, DV2, HMV, and XCV, the qPCR data showed that they were present in multiple mosquito species or biotype, while the metagenomic data suggested these viruses were only present in one species or biotype (Fig. 4 and 5). Furthermore, CPV, HMV4, WMV4, and XCV seemed to be restricted to the *Culex pipiens* biotypes. In contrast, DV2 infected *Culex pipiens pipiens* as well as *Culex torrentium*, while WMV6 even crossed the mosquito genus barrier as it was detected in all sampled mosquito species. However, it should be noted that a final viral genome copy number of 10,000 (or less) in a sample is rather arbitrary, as taking the sample dilution factor into account, these samples fell outside of the reach of the qPCR standard curve. Additionally, when breaking down the positivity rates of these viruses in the mosquitoes per location, we observe that these viruses are rather locally present (see Fig. S2; Table S2). DV is mostly present in Leuven and Bertem (<5 km apart) with a few samples in Eupen and Maasmechelen; CPV is almost exclusively present in Vrasene (one sample in Maasmechelen). XCV, on the other hand, is present in almost all locations, but the infection rates for

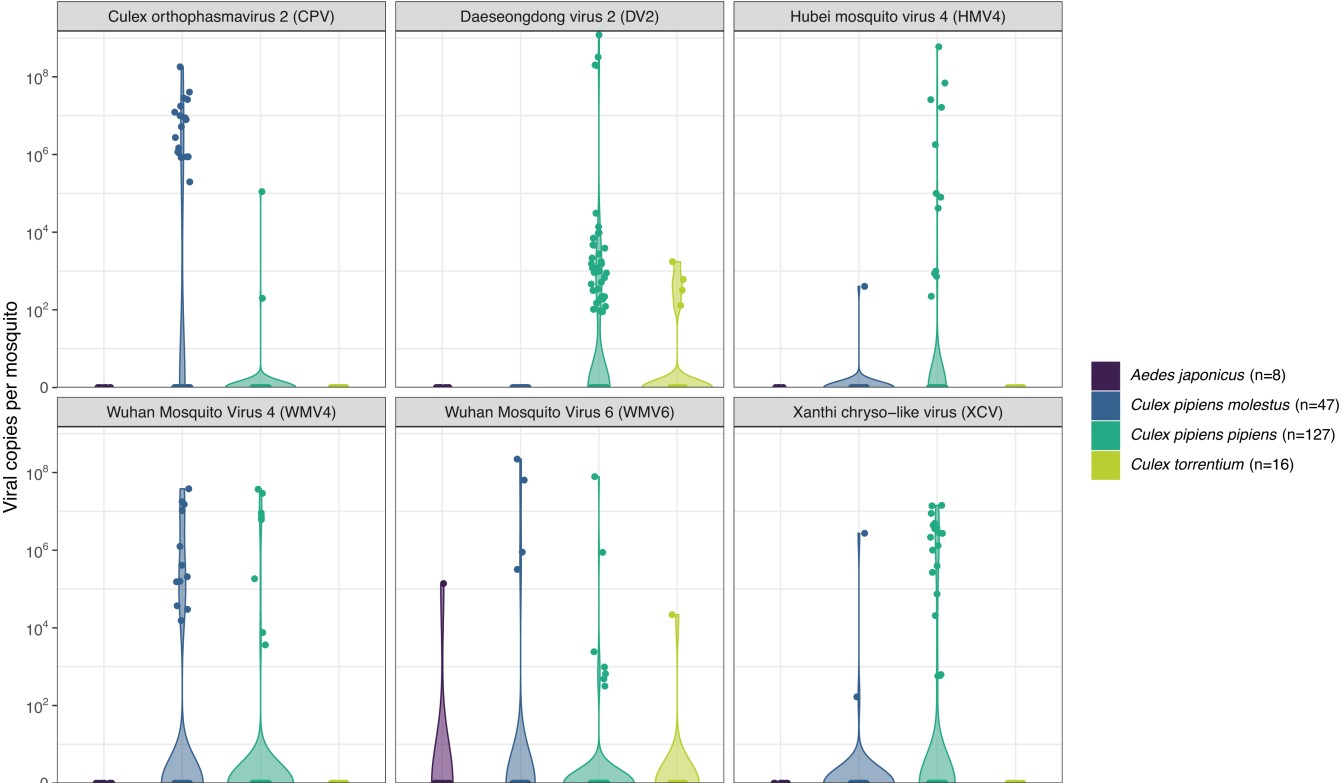

**FIG 5** Violin plots showing the quantities of selected viruses in all samples defined by qPCR. A qPCR was developed for six near-complete viral genomes in the samples (Table S1). Total viral genome copies were calculated for the whole mosquito based on the Ct-value, a standard curve, and the dilution factor of the nucleic acid extract.

*Cx. pipiens molestus* and *Cx. pipiens pipiens* are, respectively, 4.26% and 14.96% (see Table S3). This leads us to conclude that there is a lack of support for a (abundant) core virome.

## Virus phylogenetics

ORFs with complete coding sequences for the RdRP protein were extracted from all (near-)complete viral genomes with ORFfinder (23 in total). For each of the RdRP proteins, we downloaded a few close BLASTp hits with complete protein from NCBI's nr database, and also added RdRP proteins from ICTV's exemplar species for each detected viral family or order (downloaded from GenBank). Next, we performed the phylogenetic analyses with MAFFT and IQ-TREE (49, 51).

### Negevirus

Negeviruses were first described in 2013 as ISVs isolated from mosquitoes and phlebotomine sandflies (57). Although in the following years many more "negeviruses" were discovered in mostly mosquitoes, this taxon is not (yet) recognized by ICTV. Additionally, negeviruses have been found to reduce the replication of alphaviruses *in vitro* (7), making them interesting candidates to control arbovirus transmission with ISVs. Here, we found two distinct negeviruses which were both very closely related to viruses isolated from *Culex* mosquitoes from South Korea and Portugal, respectively (99% AAI with *Culex negev*-like virus 1 and 100% AAI with Negev-like virus #174, respectively; Fig. 6A; Fig. S3A), indicating a widespread occurrence.

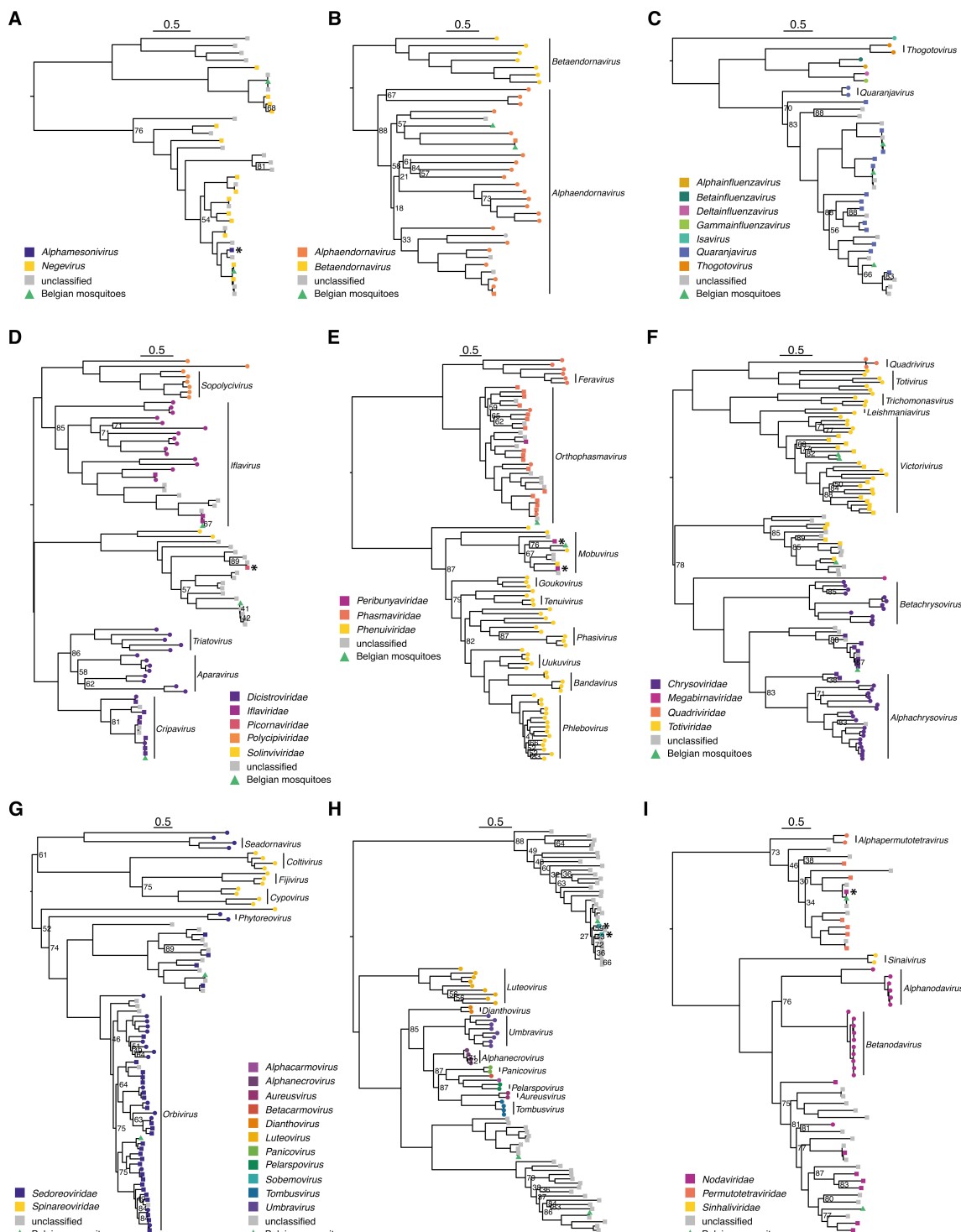

**FIG 6** Maximum likelihood, midpoint-rooted phylogenetic trees of the RdRP protein from (near-)complete genomes. Phylogenetic trees of the (A) negeviruses (unrecognized by ICTV); (B) *Endornaviridae*, (C) *Orthomyxoviridae*, (D) *Picornavirales*;, (E) *Bunyavirales*, (F) *Ghabrivirales*, (G) *Reovirales*, (H) *Tombusviridae*, (I) *Nodamuvirales,* and *Permutotetraviridae*. Next to the nodes, only bootstrap support values below 90 are shown. Viruses identified in this study are shown with a green triangle, otherwise, tips are colored by viral family (-viridae) or genus (-virus), and the shape of the tip indicates the classification source (circle: ICTV, square: NCBI, triangle: present study; ICTV was always prioritized over NCBI classification). Vertical bars with annotation represent the recognized viral genera by ICTV. Viruses wrongly classified or with outdated classification in the NCBI database are flagged with an asterisk. The scale bars indicate the number of amino acids substitutions per site.

## *Orthomyxoviridae*

To construct the phylogenetic tree of the *Orthomyxoviridae*, we focused on the PB1 segment of the RdRP complex. In our samples, we found three distinct viruses related to the genus *Quaranjavirus*, which belongs to the *Orthomyxoviridae* family (Fig. 6C; Fig. S3C). Quaranjaviruses predominantly infect arthropods and birds, and they have been associated with mass avian mortality (58). Interestingly, we detected a co-infection with WMV4 and WMV6 in five samples (see Fig. 4). Known viruses of this genus typically contain six to seven segments in databases; however, it has been proposed that they could have eight segments like the other members of the *Orthomyxoviridae* (13). As these orthomyxoviruses were among the most prevalent viruses in our samples, we could perform a co-occurrence analysis (13). A contig length corrected correlation analysis of the RdRP segments in combination with manual curation of the results enabled us to distinguish eight segments for all three orthomyxoviruses (Fig. S4). In contrast to WMV4 and WMV6, we were not able to confirm the correctness of the eighth segment for the third orthomyxovirus because there were no samples with this virus present in the SRA database and the similarity of sequences in GenBank to our own sequences was too low.

## *Picornavirales*

We identified three complete genomes from viruses belonging to the *Picornavirales* order (Fig. 6D). All three were assigned to different families, i.e., the *Iflaviridae*, *Dicistroviridae,* and the *Solinviviridae,* and are closely related to known viruses (RdRP AAI of at least 94%). All these viral families exclusively infect arthropods. First, we detected *Culex iflavi*-like virus 4, an iflavirus, that was previously identified in *Culex* mosquitoes from California and was also found before in Leuven (Belgium) in a pool of *Culex* mosquitoes (59, 60). Secondly, the dicistrovirus *Rhopalosiphum padi virus*, found in a *Culex pipiens pipiens* specimen, falls within the genus *Cripavirus* that contains multiple viruses pathogenic for insects. This virus is mostly found in aphids of the *Rhopalosiphum* and *Schizaphis* families (61), but a close relative has been described before in *Culex* mosquitoes (see Fig. S3D) (59). Furthermore, we found a third virus (94.6% AAI to Yongsan picorna-like virus 2), related to the *Solinviviridae* of which the reference species infects ants. Nonetheless, related unclassified virus sequences are derived from a large variety of insects and other arthropods (62).

## *Bunyavirales*

In the order of the *Bunyavirales*, which mainly contains vector-borne viruses, we found two viruses belonging to the *Phenuiviridae* and the *Phasmaviridae* (Fig. 6E; Fig. S3E). A novel phenuivirus (only found in the *Aedes japonicus* samples) was distantly related to *Narangue mobuvirus* (45% AAI). This virus was found previously in *Mansonia* mosquitoes in Colombia. Furthermore, we detected a phasmavirus with high similarity to *Culex orthophasmavirus* (92.3% AAI). Both detected viruses seem to be insect-specific as they are not closely related to any arbovirus in the *Bunyavirales* order.

## *Ghabrivirales*

The *Ghabrivirales* order harbors dsRNA viruses which mainly infect fungi, plants and protozoa. We found two viruses belonging to the genus *Victorivirus* that likely infect fungi in the mosquito. Nevertheless, increasing evidence suggests that members of the *Ghabrivirales* might also infect insects (63). In fact, in our phylogenetic tree of the *Ghabrivirales,* we observed, apart from the established families and genera by ICTV, two delimited clades that contain two viruses discovered in the present study and viruses from other insect metagenomes (Fig. 6F; Fig. S3F). One of these clades, harboring the widespread Xanthi chryso-like virus, falls within the *Chrysoviridae* and might be assigned as a new genus within this family. On the other hand, the second insect-specific *Ghabrivirales* clade forms a putative new viral family. In this group, mostly viruses

sequenced from mosquitoes were found and described to be "toti-like," although in this tree, they fall outside of the *Totiviridae* family. In our data set, we identified one virus in this putative new family, which is related to *Culex vishnui* subgroup totivirus (70.2% AAI), a virus found in *Culex vishnui* mosquitoes from Japan (64).

## Reovirales

Within the *Reovirales*, we selected the ICTV exemplar species of each genus that can infect invertebrates. We detected two new viruses, *Culex* orbivirus 1 and Cuseli virus 1. *Culex* orbivirus 1 was closely related to Corriparta virus in the *Orbivirus* genus (see Fig. 6G and Fig. S3G). Corriparta virus is an arbovirus discovered in 1960 in Australia and serological evidence indicates that Corriparta virus can infect humans, although no disease symptoms have been observed (65). Cuseli virus 1 falls within a clade of viruses that are not classified by ICTV. Most of the viruses in this clade were sequenced from mosquitoes and insects in general.

## Tombusviridae

Like the members of the *Ghabrivirales*, the *Tombusviridae* are mostly known for infecting plants. However, we found three viruses (Marma virus: 100% AAI; *Culex*-associated tombus-like virus: 98.5% AAI; Hubei mosquito virus 4: 96% AAI) in the Belgian mosquitoes that are related to members of the *Tombusviridae* family. Again, separate clades (potential novel genera) were formed with only insect viruses that are currently unclassified (Fig. 6H; Fig. S3H). Interestingly, some of these insect viruses are bipartite segmented viruses (13; L. De Coninck, C. Shi, and J. Matthijnssens, unpublished data), which deviates from the general assumption that genomes of members of the *Tombusviridae* are not segmented. The three *Tombusviridae* viruses detected in our sampling were all previously reported to occur in *Culex* mosquitoes from the USA, China, and Europe (13, 59, 66).

## Nodamuvirales and Permutotetraviridae

The *Nodamuvirales* order comprises the *Nodaviridae* and *Sinhaliviridae* families, both infect invertebrates, while the *Nodaviridae* family also has members that infect vertebrates. Although the *Permutotetraviridae*, an invertebrate-infecting family, is officially not classified into an order, class, or even phylum, their RdRP aligns well with those of the *Nodamuvirales*. Therefore, we included the *Permutotetraviridae* in the *Nodamuvirales* phylogenetic tree construction. Within the *Nodaviridae*, a large, diverse, unclassified cluster of insect-related viruses may represent a new genus to which our novel *Culex* noda-like virus 2 belongs (43% AAI to Hubei orthoptera virus 4). The *Permutotetraviridae* only have two officially recognized species, *Euprosterna elaeasa virus* and *Thosea asigna virus* (both isolated from the *Limacodidae* insect family), but many more viruses from insects have been discovered that fit in this family. An identical virus to Daeseongdong virus 2 (100% AAI), which has first been discovered in South-Korea (67), was present in our samples and belongs to the *Permutotetraviridae* (Fig. 6I; Fig. S3I).

## Fungal viruses

The *Endornaviridae* family infects fungi, plants, and protists. In our study, we found two known alphaendornaviruses, Tvarminne alphaendornavirus (97.6% AAI) and Hallsjon virus (98% AAI; partial RdRP not shown in Fig. 6B; Fig. S3B), most likely infecting fungi inside the mosquito (Fig. 6B; Fig. S3B). These two viruses were always found together in samples along with two totiviruses belonging to the *Victorivirus* genus (Fig. 6F). Furthermore, in these samples, a high number of fungal reads was also present. Although we found many other viruses belonging to viral families thought to exclusively infect fungi (see above), we hypothesize that only these four viruses truly infect fungi in the mosquitoes.

## Phageome and *Wolbachia* analysis

The phageome of mosquitoes is often overlooked in metagenomics studies, partly because most studies only employ an RNA sequencing strategy. Using Virsorter2 (54) and CheckV (37), we could estimate which assembled contigs were likely to be a bacteriophage and how complete those genomes were, respectively. Out of the combined Virsorter2 and CheckV results, we removed the eukaryotic virus genomes and we only considered phage genomes with a completeness estimation of more than 20%. We found seven phage contigs corresponding to four different bacteriophage species, including a complete *Microviridae* phage and three *Caudoviricetes* phages (data not shown). One of the *Caudoviricetes* phages is *Wolbachia* phage WO, which is a lysogenic phage that infects the intracellular *Wolbachia* bacterium. Lysogenic phages can incorporate their genome into the host cell's DNA to become a prophage. The *Wolbachia* genome contains five such prophage regions of phage WO (68). When blasting the phage WO contigs, a good overlap with the established prophage regions of the *w*Pip genome in GenBank was observed (Fig. 7A for a representative example and Fig. S5). In order to determine in which samples we could find phage particles and in which we only sequenced the prophage regions of the *Wolbachia* genome, we divided the average sequencing depth of the GenBank prophage regions by the average sequencing depth of the rest of the genome (excluding two rRNA gene regions). This resulted in a ratio of which we hypothesized would be much larger for samples with real phage WO particles, and we would see a clear separation between those samples and the samples with only prophage sequences. For one sample (MEMO043), it was clear that there must have been actual phage WO particles, while for the other samples, there seemed to be a low-level expression of phage WO as the coverage of prophage regions was only slightly higher (Fig. 7B; Fig. S5). Nevertheless, this low-level expression could be the result of active transcription of prophage WO-encoded accessory genes that are beneficial for the bacterium, like cytoplasmic incompatibility (*cif*) and male killing (*wmk*) genes (17, 69), rather than the induction of the lytic lifecycle of phage WO.

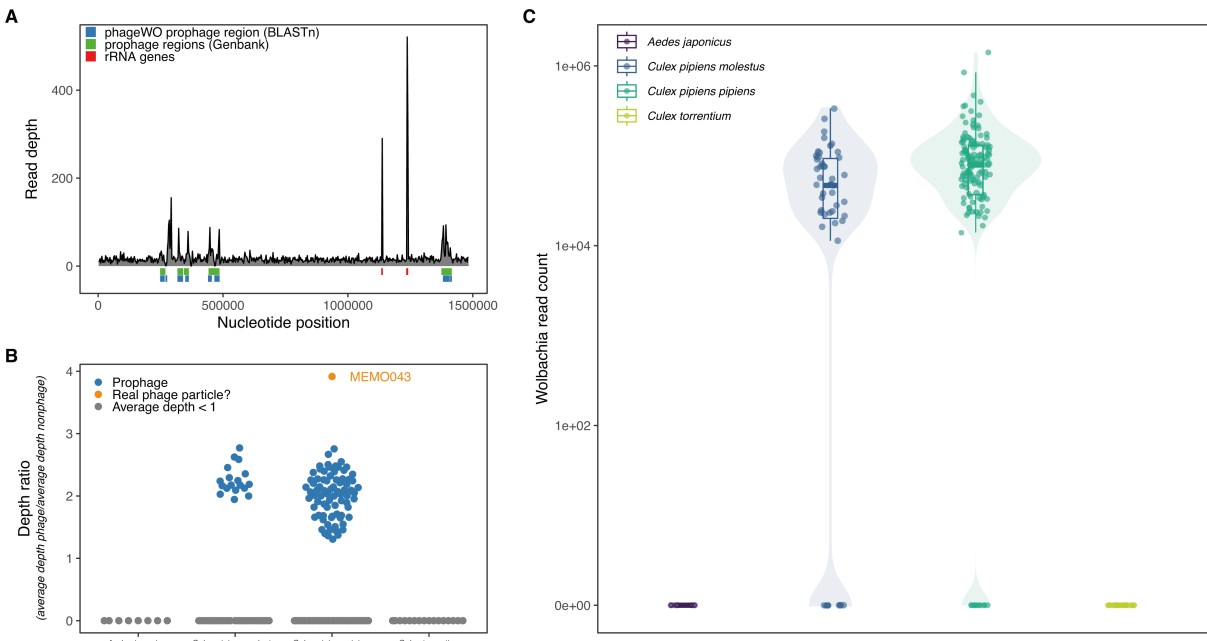

**FIG 7** *Wolbachia* prevalence estimation and phage WO prophage determination. (A) Coverage plot of the *Wolbachia* genome for sample MEMO050. The phage WO prophage regions are shown in green, together with the blastn-identified regions from the assembled phage WO contigs (blue) and rRNA genes (red). (B) Identification of real phage WO particles by dividing the average depth of the prophage regions by the average depth of the non-prophage bacterial genome regions. Samples that had an overall average sequencing depth of the Wolbachia genome lower than 1 were not considered in this analysis. (C) The number of mapped reads to the *Wolbachia* genome strain *w*Pip (AM999887.1) for each sample is shown, if the total horizontal coverage was larger than 5%.

In addition, *Wolbachia* has been shown to influence the transmission of arboviruses in mosquitoes (70–73). Therefore, we looked further into the prevalence of *Wolbachia* (sequencing reads) in our samples and found that it was present in 83% and 92.1% of *Culex pipiens molestus* and *Culex pipiens pipiens*, respectively (Fig. 7C). In contrast, none of the *Aedes japonicus* and *Culex torrentium* samples were positive for this bacterium. Similar results for *Culex pipiens pipiens* and *Culex torrentium* have been previously reported in Germany by Leggewie et al. (23). The high proportion of *Culex pipiens* mosquitoes infected with *Wolbachia* could explain why we did not find many viruses in our database, as it has been shown that *Wolbachia* infection protects insects from viral infections in general (74, 75). Therefore, we checked if there was a negative correlation between the number of viral reads versus the genome coverage of *Wolbachia* *w*Pip strain per sample. However, we were not able to identify such a correlation (see Fig. S6).

Finally, we were able to identify the *Wolbachia* pWCP plasmid (see Fig. S7). This plasmid was first discovered in *Culex pipiens* mosquitoes from France (76) and was later confirmed in a diverse set of *Culex pipiens* samples from across the world (77). It was shown that this plasmid was widely distributed and highly conserved across *Culex pipiens* mosquitoes, which we confirmed in both *Culex pipiens pipiens* and *Culex pipiens molestus*. A search in our data for other possible plasmids with geNomad (78) did not yield any results.

## DISCUSSION

In the present study, single mosquito metagenomics were used to comprehensively describe the mosquito virome of native and invasive mosquitoes in Belgium. This approach has several advantages, allowing for (i) an accurate host species determination by pairwise SNP distance estimation and unsupervised clustering from host NGS reads, and (ii) the determination of the prevalence rate of viruses and bacteria (e.g., *Wolbachia*) in mosquito populations (13). Overall, viruses seemed absent in a large proportion of the mosquitoes analyzed (Fig. 1C), which was surprising as previous studies on individual mosquitoes did not find such a high number of virus-negative samples (9, 13). A recent country-wide study on individual mosquitoes from China, however, found that *Culex pipiens,* on average, harbored 2.34 (±1.52 SD) virus species per individual (*n* = 438) (14), which is within the range of our observations. A possible explanation for these differences could be the overall higher mean temperatures in Guadeloupe and California compared to Belgium, as it is known that temperature can have an influence on virus replication and infection (79). This hypothesis is also reinforced by the observation of Feng et al. (10), which revealed that *Culex pipiens* mosquitoes from the same region have a remarkably lower viral abundance in colder months compared to warmer ones.

Nevertheless, the detected viruses were highly diverse, and included a close relative to Corriparta virus which is capable of infecting humans (65). Most other viruses were also RNA viruses, which is in line with similar studies on insect viromes (2, 9, 80). In congruence with our observations, Abbo et al. found highly diverse viromes in *Aedes japonicus* (80). In addition, *Aedes japonicus* mosquitoes had a more diverse virome than mosquitoes from the *Culex* genus, as shown with different alpha diversity metrics (Fig. 3A). This seems to be a recurring observation in direct comparisons between *Aedes* and *Culex* samples, where *Aedes* mosquitoes have a higher viral diversity (6, 9). Furthermore, the virome composition also differs significantly between these latter mosquito genera (Fig. 3B and C and Fig. 4; Fig. S2), underlining that the viral diversity is more driven by the host species than by the collection location, thereby hinting on a specific core virome for each investigated mosquito species (9, 60). However, weaknesses of this study include the low virus-positive sample sizes for *Aedes japonicus* (*n* = 8) and *Culex torrentium* (*n* = 3), and additionally the single collection location of the *Aedes japonicus* samples (i.e., Natoye) without any other mosquito species present there. Therefore, differences in the virome between the *Culex* and *Aedes* samples might reflect the collection site and not the mosquito species. However, as mentioned before, other studies have observed similar differences between these two mosquito genera (9, 81). In addition, also the low

number of viruses per individual warrants a cautious interpretation of the beta diversity analyses.

A "core virome" in the sense of a select group of (insect-specific) viruses that are present in almost all mosquitoes of the same species did not apply in our study, considering that we observed only a few viruses present in a minority of samples from the same species. The "mosquito core virome" concept was originally coined after a virome study using *Aedes aegypti* and *Culex quinquefasciatus* mosquitoes captured in Guadeloupe, an archipelago situated where the Atlantic Ocean meets the Caribbean Sea (9). As the influx of mosquitoes with new ISVs into the population might be more limited on islands than on mainland, the observation of this core virome might have been enhanced due to the continuous replication of the same ISVs in this confined mosquito population. Furthermore, there are also geographical differences, where Belgium is more urbanized and, therefore, most likely has a less continuous, more disrupted ecosystem which, we hypothesize, can have an impact on the mosquito virome. Interestingly, a virome study of African and European *Culex pipiens* mosquitoes with RT-qPCR (after initial metagenomics on mosquito pools) also did not find a core virome when comparing individual samples (82). Ultimately, in our judgment, more single mosquito metagenomics studies on larger geographical scales are necessary to confirm or reject the "mosquito core virome" concept.

Due to the broad diversity of viral genomes and the nature of metagenomic experiments, it is difficult to undoubtedly infer the host of the discovered viruses. Despite these inherent difficulties, we showed that there are many viruses in mosquitoes that are related to viruses in established viral families assumed to strictly infect plants or fungi (Fig. 6F and H). By extension, this has also been observed in several other studies on a variety of insects (2, 9). Due to the growing number of reports of such viruses in insect metagenomic studies, we believe that these viruses also truly infect these insects and are not merely "passerby" viruses that originated from plants as a food source (e.g., nectar) or from fungi that infect the insect. Additional information pointing in this direction is the discovery that viruses in insects closely related to the *Tombusviridae* can have bisegmented genomes, contrasting with tombusviruses found in plants which have a single RNA strand as genome (unpublished data and genomes generated by Batson et al. [13]). However, infection experiments in insect cell lines as well as insect models could help to validate this hypothesis. Furthermore, in addition to the classification and host inference difficulties related to novel virus identification, we noticed that information about (insect) viral genomes in the NCBI database can occasionally be incorrect or outdated. For example, viral genomes are occasionally misclassified by the submitters, e.g., *Alphamesonivirus* in the Negevirus family, a virus designated as *Picornaviridae* in a clade with unclassified and *Solinviviridae* sequences, *Peribunyaviridae* classifications in *Phasmaviridae* as well as *Phenuiviridae* clades, and finally, a virus classified as *Nodaviridae* in a clear *Permutotetraviridae* clade (Fig. 6A, D, E, and I, respectively). This can impede a correct analysis of metagenomic studies and can potentially lead to more misclassifications for novel viral genomes in the NCBI databases if, for instance, large metagenomic studies automatically assign the taxonomic classification of their novel viruses by a similarity-based approach and submit those genomes without performing rigorous checks. This stresses the need for a peer-reviewed system to efficiently change and correct information about biological sequences on the NCBI servers.

Finally, we aimed to address a major understudied part of the mosquito virome: the phageome, or the collection of viruses that infect bacteria. However, we did not find large bacteriophage communities in our samples in contrast to Shi et al. (9). The latter study, however, was performed with older bacteriophage identification tools and on short sequences (mostly <1,500 bp), which seriously hampers a correct identification of bacteriophages. Therefore, the results of Shi et al. do not reflect the current state-of-the-art in bacteriophage research and at present should be considered with caution. A potential explanation for the lack of phages could be that mosquitoes might not have a long history of co-evolution with their bacteriome, as the bacteria in the mosquito

are mostly obtained from the environment (4, 83), thus giving little time to establish a complex relationship between the mosquito host on one side and the bacteria and their phages on the other side. However, the *Wolbachia pipientis* bacterium is an exception, as it is transmitted from parent to offspring because it resides intracellularly in the reproductive system. We found two phage WO contigs in our data set, although we cannot fully exclude that these are parts of prophage regions in the *Wolbachia* genome. These prophage WO regions, which can act as a mobile element for horizontal gene transfer, harbor genes that are important to induce cytoplasmic incompatibility (CI) (17, 76). CI is the inability of an infected male and an uninfected female insect to reproduce, and consequently is an important mechanism for *Wolbachia* to spread in the population. *Wolbachia* has also been shown to influence the viral transmission of several arboviruses (70–73); therefore, its prevalence in the mosquito population and the role of (pro)phages in *Wolbachia*'s dissemination should be studied further. Hence, bacteriophages present in mosquitoes should not be overlooked.

In conclusion, we report the lack of an abundant core virome in *Culex* mosquitoes from Belgium and propose to tread more lightly in defining mosquito core virome members and the interpretation of what a "core virome" could mean biologically. Additionally, 28 novel viruses were identified, which will contribute to our understanding of the mosquito virome and ISVs.

## ACKNOWLEDGMENTS

We would like to thank all cooperating companies for giving access to their private property during mosquito sampling surveys. In addition, we would like to thank the laboratory and technical staff of the Entomology Unit at the Institute of Tropical Medicine.

This research was supported by the Research Foundation Flanders (11L1323N [L.D.C.]) and a KU Leuven grant (C14/20/108). Part of the mosquito samples have been collected by the Institute of Tropical Medicine Antwerp (ITM) during the MEMO project (tender: CES-2016-02 Belgium), funded by the Flemish, Walloon and Brussels regional governments and the Federal Public Service (FPS) Public Health, Food Chain Safety and Environment within the framework of the Belgian national collaboration agreement in the domains of environment and health (NEHAP). Samples have been barcoded by the Barcoding Facility for Organisms and Tissues of Policy Concern (BopCo), financed by the Belgian Science Policy Office (Belspo). The computational resources and services used in this work were provided by the VSC (Flemish Supercomputer Center), funded by the Research Foundation-Flanders (FWO) and the Flemish Government.

## AUTHOR AFFILIATIONS

[1]KU Leuven, Department of Microbiology, Immunology and Transplantation, Rega Institute, Division of Clinical and Epidemiological Virology, Laboratory of Viral Metagenomics, Leuven, Belgium
[2]KU Leuven, Department of Microbiology, Immunology, & Transplantation, Rega Institute, Laboratory of Virology and Chemotherapy, Mosquito Virology Team, Leuven, Belgium
[3]Department Biomedical Sciences, The Unit of Entomology, Institute of Tropical Medicine, Antwerp, Belgium
[4]Department of Biology, Terrestrial Ecology Unit, Ghent University, Ghent, Belgium
[5]Department of Biology, Royal Museum for Central Africa (Barcoding Facility for Organisms and Tissues of Policy Concern), Tervuren, Belgium

## AUTHOR ORCIDs

Lander De Coninck  http://orcid.org/0000-0001-6847-2379
Leen Delang  https://orcid.org/0000-0002-8874-675X
Jelle Matthijnssens  http://orcid.org/0000-0003-1188-9733

## FUNDING

| Funder | Grant(s) | Author(s) |
|---|---|---|
| Fonds Wetenschappelijk Onderzoek (FWO) | 11L1323N | Lander De Coninck |
| KU Leuven | C14/20/108 | Leen Delang |
| | | Jelle Matthijnssens |

## DATA AVAILABILITY

Raw sequencing data have been made available through NCBI's Sequence Read Archive (SRA) under Bioproject PRJNA880624. Complete viral genomes were submitted to GenBank under accession numbers PP076491-PP076718. All data and code for the virome analysis have been deposited on Github (https://github.com/Matthijnssenslab/2024_mSystems_BelgianMosquitoVirome).

## ADDITIONAL FILES

The following material is available online.

### Supplemental Material

**Supplemental material (mSystems00012-24-s0001.pdf).** Supplemental figures and tables.

### Open Peer Review

**PEER REVIEW HISTORY (review-history.pdf).** An accounting of the reviewer comments and feedback.

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
