## [Reviewer comments · mSystems]

Lack of abundant core virome in *Culex* mosquitoes from a temperate climate region despite a mosquito species-specific virome

Lander De Coninck, Alina Soto, Lanjiao Wang, Katrien De Wolf, Nathalie Smitz, Isra Deblauwe, Karelle Mbigba Donfack, Ruth Muller, Leen Delang, and Jelle Matthijnssens

Corresponding Author(s): Jelle Matthijnssens, Katholieke Universiteit Leuven

Review Timeline:

Submission Date:	January 4, 2024
Editorial Decision:	February 22, 2024
Revision Received:	March 15, 2024
Accepted:	April 15, 2024

Editor: Seth Bordenstein

Reviewer(s): The reviewers have opted to remain anonymous.

Transaction Report:

DOI: <https://doi.org/10.1128/mSystems.00012-24>

Re: mSystems00012-24 (Lack of abundant core virome in Culex mosquitoes from a temperate climate region despite a mosquito species-specific virome)

Dear Prof. Jelle Matthijnssens:

Thank you for the opportunity to review your submission to mSystems. Two expert reviewers both appreciated the value of discovering new viruses in the system, and one reviewer had concerns about the analyses that I would like you to consider further with my recommendation of revise and resubmit. Below you will find instructions from the mSystems editorial office and the reviewer comments.

Please return the revised manuscript within 60 days; if you cannot complete the modification within this time period, please contact me. If you do not wish to modify the manuscript and prefer to submit it to another journal, notify me immediately so that the manuscript may be formally withdrawn from consideration by mSystems.

Revision Guidelines

Sincerely,
Seth Bordenstein
Editor
mSystems

Reviewer #1 (Comments for the Author):

Review of Manuscript: "Lack of abundant core virome in Culex mosquitoes from a temperate climate region despite a mosquito species-specific virome"

I had the privilege of reviewing the manuscript, and I must commend the authors for their excellent work. The study delves into

the intricate world of insect-specific viruses (ISVs) within arthropod-associated microbial communities, offering valuable insights into the virome of native and invasive mosquito species in Belgium.

Strengths:

1. **Clarity and Conciseness:** The manuscript is exceptionally well-written, with a clear and concise presentation of methods and results. The authors effectively communicate complex concepts, ensuring accessibility for a broad readership.
2. **Innovative Approach:** The utilization of single mosquito metagenomics is a commendable approach that allows for accurate host species determination and the identification of novel viruses. This innovative methodology significantly contributes to the field.
3. **Novel Discoveries:** The identification of 45 viruses, including 28 novel ones, enriches our understanding of the mosquito virome and ISVs. The species-specific nature of the mosquito virome, irrespective of geographical location, is a noteworthy finding.

Constructive Feedback:

1. **Introduction Enhancement:** I would recommend incorporating information about the World Mosquito Program (WMP) in the introduction. This addition would contextualize the study within broader mosquito control efforts and potentially enhance its relevance to a wider audience.
2. **Bacterial Genomes Exploration:** Consider expanding the exploration of bacterial genomes beyond Wolbachia. Providing additional context and insights into other bacterial elements could further enrich the study.
3. **Discussion Expansion:** The discussion section could be extended to delve deeper into the implications of the findings. This may include further interpretation of results, addressing study limitations, and proposing avenues for future research. Connecting the study's outcomes to practical applications in mosquito control or disease prevention would strengthen the impact of the research.

Conclusion:

In conclusion, this manuscript represents a significant contribution to the field of mosquito virology. The meticulous methodology, coupled with novel discoveries, positions this work as a valuable reference for researchers and practitioners alike. I commend the authors on their exemplary work and look forward to witnessing the continued impact of their research in the scientific community.

Reviewer #2 (Comments for the Author):

The manuscript by de Coninck et al. describes the diversity of viruses found in individual mosquitoes collected in different sites in Belgium. Mosquitoes belonged to three species, including two species of the *Culex* genus and one *Aedes* species. The authors screened 190 mosquito individuals in total using metagenomics. Mosquitoes were also screened for the presence of six viruses by RT-PCR. Moreover, the authors determined Wolbachia prevalence from the sequencing data obtained with metagenomics. The authors present analyses of the virus communities and phylogenies of the viruses found in their mosquito collection, including the description of new species.

The main result of the manuscript, as defined by the title and the abstract, is the lack of a clear support for the existence of a core virome in the studied populations. This claim mainly derives from the limited diversity overlap in the virome of individual mosquitoes observed with metagenomics. I see two problems in the manuscript suggesting that the results do not robustly support this claim.

First, assessing a core virome requires a diagnostic test allowing to robustly infer presence/absence of viruses in the samples. The approach used by the author does not seem to fully comply with this requirement. No virus was found in a large number of mosquitoes (61 %). I have not found an explanation for this result or a comparison with other studies. Few studies have analyzed individual mosquitoes with metagenomics but at least two of them (including a study from the same group) did not find such a high proportion of negative samples (references 8 and 12 in the manuscript). The low number of virus-positive samples thus suggest a sensibility problem (either due to the method or the sample quality).

Secondly, there seems to be contradictory results between the metagenomics and RT-PCR approaches. The authors describe that the RT-PCR approach, an approach more sensitive than metagenomics, allows to detect six viruses in more individuals than those found with metagenomics (lines 347-349). The viruses in the RT-PCR analysis seem to have been selected based on their prevalence in the two biotypes of *Culex pipiens*. However, the infection rates in those biotypes obtained with RT-PCR are not clearly provided in the Results section. The number of individuals with a positive detection cannot be easily determined from Figure 5 but there seems to be a large number of individuals of each biotype infected by certain viruses (e.g., see results for Daseongdong virus). If this is the case, there seems to be some viruses that are relatively prevalent thus questioning the claim of a lack of a core virome.

Another main claim of the paper is that the results show an influence of the mosquito species on the virome. The problem I see with this claim is that the sample sizes are too low to be able to do a robust analysis. The number of virus-positive individuals obtained with metagenomics is extremely low for two out of the three species (three and eight individuals for *Culex torrentium* and *Aedes japonicus* respectively). These sample sizes do not allow for a robust comparative analysis of the influence of mosquito species on the virome or on the existence of a core virome in those mosquito species.

Other comments:

There are several features of the study that limit the robustness of the alpha- and beta-diversity analyses. These features include the limited number of virus-positive individuals for certain mosquito species (see comment above) and the limited number of viruses detected per virus-positive individuals (only one virus was detected in most virus-positive individuals). Those analyses are sensitive by incomplete species identification (e.g., see Beck et al., 2010 *Methods in Ecology and Evolution*). Moreover, there are nested factors that hamper some analyses. For example, one site provided all *Aedes* individuals and none from the other species (Natoye site if I have well understood Figure 1A; colors are not easily distinguishable). That is, differences between *Aedes* individuals and the other species may be due to the mosquito species or to specificities of the collection site. Overall, the analyses of species diversity and structure do not seem to be enough solid to allow conclusions.

The authors have used different traps that collect individuals in different life stages (e.g., gravid females versus non-gravid females). Such differences could heavily influence results. The authors do not provide data on the life stage of individuals that could help to study the influence of that factor.

Lines 25-26: "Contrary to expectations..." Previous work should have led to expect the results. A study on one of the mosquito species (*Culex pipiens*) in the manuscript has shown that most individuals do not share a set of viruses commonly found in populations of that mosquito in different countries in Africa and Europe (Gil et al, 2023 *Virus Evolution*). That is, the study did not detect a core virome when comparing individuals, just like in the manuscript.

Lines 26-27. I agree with the authors that studies on the core virome should define the population level at which the core is analyzed (i.e. between individuals of a population, between populations in a country, between countries or continents, etc.). The authors clearly provide their definition of a core virome ("a select group of (insect-specific) viruses that are present in almost all mosquitoes of the same species"; lines 559-560). However, previous work has already shown that a core virome can be observed if the population scale changes. Several articles have shown that a set of viruses is found associated to populations of a given mosquito species over large sections of the mosquito geographical range, including mosquito species targeted in the manuscript (e.g., Shi et al, 2020; Gil et al, 2023; Moonen et al 2023). The fact that the authors have not observed a core virome at the studied spatial level (i.e., populations situated less than 150 km apart) does not mean that a core virome takes place at a higher level, for example between populations in different continents. The authors do not discuss this point despite a study clearly showing the influence of scale on the virome in the mosquito species with more individuals in the manuscript (Gil et al, 2023).

Line 68. A quantitative analysis of publications on mosquito viromes showed that the mosquito genus with most publications is the *Culex* genus (Moonen et al, 2023).

Lines 562-564 "The "core virome" concept was originally coined after a virome study using *Aedes aegypti* and *Culex quinquefasciatus* mosquitoes captured in Guadeloupe,...". In fact, the term "core virome" had already been used in another study (Broecker et al, 2017 *Gut Microbes*). Moreover, the use of the term "core" was widespread in community ecology long before the paper cited by the authors.

Lines 568-569. The authors do not provide a reference and there is a study that obtained results diverging from this sentence (reference 12).

Lines 577-580. There is no need to invoke the Amplification effect hypothesis. Human activity almost always leads to drops in biodiversity, including microorganism diversity. I find difficult to follow this section of the discussion.

Lines 611-612. The reason behind the limited number of bacteriophage sequences seems not the more plausible and the authors do not provide references supporting it. A previous work from the same group found a dominance of bacteriophage sequences in *Culex* mosquitoes (reference 8). Thus, an alternative hypothesis is a sensibility problem.

- few data on the *Culex* virome: to nuance: more publications on *Culex* viruses than the other genus (see Moonen 2023 *One Health*)

Dear editors,

Please find herewith our revised manuscript, entitled “Lack of abundant core virome in *Culex* mosquitoes from a temperate climate region despite a mosquito species-specific virome,” authored by Lander De Coninck and co-authors.

Thank you for giving us the opportunity to respond to the Reviewer’s comments and, hence, improve the quality of our paper. We hope this revised manuscript is now acceptable for publication in mSystems.

We responded to the comments of the reviewers as follows:

Reviewer #1 (Comments for the Author):

Review of Manuscript: "Lack of abundant core virome in *Culex* mosquitoes from a temperate climate region despite a mosquito species-specific virome"

I had the privilege of reviewing the manuscript, and I must commend the authors for their excellent work. The study delves into the intricate world of insect-specific viruses (ISVs) within arthropod-associated microbial communities, offering valuable insights into the virome of native and invasive mosquito species in Belgium.

Strengths:

1. Clarity and Conciseness: The manuscript is exceptionally well-written, with a clear and concise presentation of methods and results. The authors effectively communicate complex concepts, ensuring accessibility for a broad readership.
2. Innovative Approach: The utilization of single mosquito metagenomics is a commendable approach that allows for accurate host species determination and the identification of novel viruses. This innovative methodology significantly contributes to the field.
3. Novel Discoveries: The identification of 45 viruses, including 28 novel ones, enriches our understanding of the mosquito virome and ISVs. The species-specific nature of the mosquito virome, irrespective of geographical location, is a noteworthy finding.

We thank reviewer #1 for their compliments and constructive feedback, our answers to their requests can be found below.

Constructive Feedback:

1. Introduction Enhancement: I would recommend incorporating information about the World Mosquito Program (WMP) in the introduction. This addition would contextualize the study within broader mosquito control efforts and potentially enhance its relevance to a wider audience.

We mentioned the WMP and its goals now on lines 77-81 in the introduction.

2. Bacterial Genomes Exploration: Consider expanding the exploration of bacterial genomes beyond *Wolbachia*. Providing additional context and insights into other bacterial elements could further enrich the study.

We agree that a broader view on the bacteriome would enrich the study, however our data does not easily allow this as we specifically enriched for viruses, which severely diminishes and biases bacterial diversity. We therefore only focused on *Wolbachia* which is shown to influence vector competence and which we knew was naturally present in the *Culex* mosquito population in Belgium.

3. Discussion Expansion: The discussion section could be extended to delve deeper into the implications of the findings. This may include further interpretation of results, addressing study limitations, and proposing avenues for future research. Connecting the study's outcomes to practical applications in mosquito control or disease prevention would strengthen the impact of the research.

We further improved our discussion based on the suggestions of both reviewers:

Lines 505-514: we added a comparison with other studies.

Lines 526-533: we added the low sample sizes for *Aedes japonicus* and *Culex torrentium* and the low viruses identified as a weakness for the beta diversity analyses.

Lines 545-549: Provided extra reference and comparison to a lack of core virome on the individual mosquito level

Conclusion:

In conclusion, this manuscript represents a significant contribution to the field of mosquito virology. The meticulous methodology, coupled with novel discoveries, positions this work as a valuable reference for researchers and practitioners alike. I commend the authors on their exemplary work and look forward to witnessing the continued impact of their research in the scientific community.

Reviewer #2 (Comments for the Author):

The manuscript by de Coninck et al. describes the diversity of viruses found in individual mosquitoes collected in different sites in Belgium. Mosquitoes belonged to three species, including two species of the *Culex* genus and one *Aedes* species. The authors screened 190 mosquito individuals in total using metagenomics. Mosquitoes were also screened for the presence of six viruses by RT-PCR. Moreover, the authors determined *Wolbachia* prevalence from the sequencing data obtained with metagenomics. The authors present analyses of the virus communities and phylogenies of the viruses found in their mosquito collection, including the description of new species.

The main result of the manuscript, as defined by the title and the abstract, is the lack of a clear support for the existence of a core virome in the studied populations. This claim mainly derives from the limited diversity overlap in the virome of individual mosquitoes observed with metagenomics. I see two problems in the manuscript suggesting that the results do not robustly support this claim.

We thank reviewer #2 for their critical reading of our manuscript and their insights. Our responses can be found below:

First, assessing a core virome requires a diagnostic test allowing to robustly infer presence/absence of viruses in the samples. The approach used by the author does not seem to fully comply with this requirement. No virus was found in a large number of mosquitoes (61 %). I have not found an explanation for this result or a comparison with other studies. Few studies have analyzed individual mosquitoes with metagenomics but at least two of them (including a study from the same group) did not find such a high proportion of negative samples (references 8 and 12 in the manuscript). The low number of virus-positive samples thus suggest a sensibility problem (either due to the method or the sample quality).

In agreement with reviewer #2, we were also surprised to see that we only had such a low number of virus positive samples. However, we believe this is not a result of the method and/or low sample quality:

- 1) Our method has been widely used to detect viruses in different sample types (human gut, insects [including mosquitoes, bees and blackflies], bats, plants, etc.). In addition, we also optimized our method in a small pilot study (not published) to recover the virus read yield for mosquitoes (eg. different homogenization speeds, concentration of viruses by centrifugation with molecular weight filters). The method used here, and in Shi *et al.* (2019), was still the most performant.
- 2) As mentioned by reviewer #2, few studies have performed single mosquito metagenomics so direct comparisons are difficult because other *Culex* virome studies in a temperate climate region only used pools. Pools can mask the large amount of mosquitoes without viruses. The studies we can compare to: have a markedly warmer climate (Shi *et al.* 2019: Guadeloupe; Batson *et al.* 2021: California) and for the California study, samples were also collected during a period with unusually warm temperature peaks (eg. San Diego, fall 2017:

<https://weatherspark.com/h/s/1816/2017/2/Historical-Weather-Fall-2017-in-San-Diego-California-United-States>). In our eyes, further strengthening the hypothesis that higher temperatures could lead to more virus in these mosquitoes (mentioned on lines 569-575 of the original submission). Furthermore, a recent preprint on viromics of 2438 individual mosquitoes from China found that *Culex pipiens* (n=438) on average harbored 1.84 (± 1.65 SD) viruses which is within the range of our study (Pan *et al.*, 2023 *bioRxiv*, <https://doi.org/10.1101/2023.08.28.555221>). Of note, these authors did find a negative correlation between viral richness and annual mean temperature, but we have concerns regarding this result as they did not make a distinction between mosquito species, they only used the average annual temperature which might not reflect the actual temperature the mosquitoes were caught in and finally, in the low temperature range they have a lot less samples than in the high temperature range which might skew their results.

- 3) Reviewer #2's suggestion of low sample quality is hard to refute, but we stored our samples from collection to analysis on -80°C as usual for these kind of samples (and widely described in literature). Besides, for the *Aedes* mosquitoes, we did find multiple viruses in every sample suggesting that sample quality and our method were not an issue.

We added a comparison to other studies in the discussion on lines 505-514.

Secondly, there seems to be contradictory results between the metagenomics and RT-PCR approaches. The authors describe that the RT-PCR approach, an approach more sensitive than metagenomics, allows to detect six viruses in more individuals than those found with metagenomics (lines 347-349). The viruses in the RT-PCR analysis seem to have been selected based on their prevalence in the two biotypes of *Culex pipiens*. However, the infection rates in those biotypes obtained with RT-PCR are not clearly provided in the Results section. [We added a supplementary table with infection rates and a supplementary figure.] The number of individuals with a positive detection cannot be easily determined from Figure 5 but there seems to be a large number of individuals of each biotype infected by certain viruses (e.g., see results for Daseongdong virus). If this is the case, there seems to be some viruses that are relatively prevalent thus questioning the claim of a lack of a core virome.

Based on Figure 5, an argument could be made to question the lack of a core virome as some viruses (DV, CPV and XCV) seem to be present in a large number of individuals. However, based on the following arguments, we would like to better explain our arguments, for NOT claiming the presence of an "abundant" core virome. First of all, samples with a genome copy number of 10,000 or less were positive for the virus but their Ct was higher than the lowest concentration of our standard curve, making the genome copy determination not entirely accurate (we mentioned this on lines 355-358 of the original manuscript). Certainly for DV only a small number of samples had copy numbers above this threshold, DV is therefore not very abundant when it is present and this is mentioned in the title of the manuscript: "Lack of **abundant** core virome...". Secondly, primers were designed in the RdRP gene (across multiple sequences from the different samples), this gene might be more conserved across viral species. Therefore, there is also the possibility that we detected similar, but different, viral species with our RT-qPCR. Finally, our most

compelling argument to not call it a core virome comes from the observation that these viruses are quite locally present (see new supplementary tables and figure below). DV is mostly present in Leuven and Bertem (<5km apart) with a few samples in Eupen and Maasmechelen, CPV is almost exclusively present in Vrasene (1 sample in Maasmechelen). XCV on the other hand is present in almost all locations, but the infection rates for *Cx. pipiens molestus* and *Cx. pipiens pipiens* are respectively 4.26% and 14.96%. All this information leads us to conclude that there is a lack of support for an (abundant) core virome.

We added this reasoning in the Results section on lines 331-338.

Another main claim of the paper is that the results show an influence of the mosquito species on the virome. The problem I see with this claim is that the sample sizes are too low to be able to do a robust analysis. The number of virus-positive individuals obtained with metagenomics is extremely low for two out of the three species (three and eight individuals for *Culex torrentium* and *Aedes japonicus* respectively). These sample sizes do not allow for a robust comparative analysis of the influence of mosquito species on the virome or on the existence of a core virome in those mosquito species.

We agree that for *Culex torrentium* and *Aedes japonicus* virus-positive sample counts are low, however it is already known from literature that viromes seem to be species-specific. Certainly differences between *Aedes* and *Culex* have been shown before (Shi *et al.* 2019; Thongsirong *et al.*, 2021, *Sci. Rep.*; Li *et al.*, 2023, *Microbiology Spectrum*; Wang *et al.* 2024, *Microbiology Spectrum*).

We mention this now as a weakness of the study on lines 526-528.

Other comments:

There are several features of the study that limit the robustness of the alpha- and beta-diversity analyses. These features include the limited number of virus-positive individuals for certain mosquito species (see comment above) and the limited number of viruses detected per virus-positive individuals (only one virus was detected in most virus-positive individuals). Those analyses are sensitive by incomplete species identification (e.g., see Beck et al., 2010 *Methods in Ecology and Evolution*).

We acknowledge the concern of reviewer #2 about the issue of undersampling and although we cannot exclude the possibility that we have missed to identify viruses, we believe that our method is still sensitive enough to do these types of analyses (see also answers above). However, the low number of viruses per individual warrants a cautious interpretation of the NMDS and PCoA plots (added this as a weakness on lines 531-533). In addition, these analyses are also sensitive to overestimation of (virus) species. Therefore, we have chosen to be more stringent in our identification of viruses to not include genome fragments.

Moreover, there are nested factors that hamper some analyses. For example, one site provided all *Aedes* individuals and none from the other species (Natoye site if I have well understood Figure 1A; colors are not easily distinguishable) [Colors have been made more divergent in all figures]. That is, differences between *Aedes* individuals and the other species may be due to the mosquito species or to specificities of the collection site. Overall, the analyses of species diversity and structure do not seem to be enough solid to allow conclusions.

We agree and we added this as a weakness in the discussion on lines 526-528. But other studies have observed similar differences between *Aedes* and *Culex*, and between different mosquito genera in general (Shi *et al.* 2019; Thongsirong *et al.*, 2021, *Sci. Rep.*; Li *et al.*, 2023, *Microbiology Spectrum*; Wang *et al.* 2024, *Microbiology Spectrum* as a few examples).

The authors have used different traps that collect individuals in different life stages (e.g., gravid females versus non-gravid females). Such differences could heavily influence results. The authors do not provide data on the life stage of individuals that could help to study the influence of that factor.

Unfortunately, we do not have the information anymore on the gravidness of the mosquitoes (the samples were homogenized before assessing). However, we added the trap type as an extra explaining variable into the PERMANOVA analysis. This showed that trap type (and by assumption gravidness, if we consider mosquitoes caught in gravid traps to be gravid and mosquitoes caught by the other traps to be non-gravid) only had a minor influence on the virome composition (4% vs 25% for mosquito species, and 10% for collection location).

Lines 25-26: "Contrary to expectations..." Previous work should have led to expect the results. A study on one of the mosquito species (*Culex pipiens*) in the manuscript has shown that most individuals do not share a set of viruses commonly found in populations of that mosquito in different countries in Africa and Europe (Gil *et al.*,

2023 Virus Evolution). That is, the study did not detect a core virome when comparing individuals, just like in the manuscript.

Thank you for making us aware of the paper from Gil *et al.*, this study was published some time after we started this project thus our results were still contrary to “our” expectations, we have modified this on line 25 to “Contrary to **our** expectations...”. In addition, Gil *et al.* only did RT-qPCR on the individual level in combination with NGS on pools, the latter might have masked the presence of lower abundant viruses and would therefore not be identified for the subsequent RT-qPCRs. We believe that our study is complementary with Gil *et al.* and even provides stronger evidence for the lack of a core virome. We discussed the Gil *et al.* results on lines 545-547.

Lines 26-27. I agree with the authors that studies on the core virome should define the population level at which the core is analyzed (i.e. between individuals of a population, between populations in a country, between countries or continents, etc.). The authors clearly provide their definition of a core virome ("a select group of (insect-specific) viruses that are present in almost all mosquitoes of the same species"; lines 559-560). However, previous work has already shown that a core virome can be observed if the population scale changes. Several articles have shown that a set of viruses is found associated to populations of a given mosquito species over large sections of the mosquito geographical range, including mosquito species targeted in the manuscript (e.g., Shi et al, 2020; Gil et al, 2023; Moonen et al 2023).

We would like to point out that the studies mentioned by reviewer #2 are based on pools of mosquitoes which mask, as previously mentioned, the true prevalence of viruses in individual mosquitoes. In our opinion, we do not feel that viruses are part of a core virome when they are present in a minority of the individual samples across a large geographical scale (or a small one for that matter). A parallel for this reasoning can be made with eg. Influenza virus (or other human viruses), which will be present in a minority of humans across a large geographical scale, but is obviously not considered to be a human core virus. Furthermore, as mentioned above by reviewer #2, Gil *et al.* also did not find a core virome on the individual level.

The fact that the authors have not observed a core virome at the studied spatial level (i.e., populations situated less than 150 km apart) does not mean that a core virome takes place at a higher level, for example between populations in different continents. The authors do not discuss this point despite a study clearly showing the influence of scale on the virome in the mosquito species with more individuals in the manuscript (Gil et al, 2023).

We can indeed not fully exclude that a core virome does not exist on a different spatial level. It would, however, feel contradictory that on a smaller scale a core virome would not be present, while it would be on a larger spatial scale. The study referenced by reviewer #2 shows to us that when using pools on a larger geographical scale, viruses can be found that are shared, but when looked at individuals on a smaller scale, these viruses are not prevalent across the studied mosquito population. Studies on mosquito pools (Pettersen *et al.* 2019, *Viruses*; Li *et al.*, 2023, *Microbiology Spectrum*; Wang *et al.* 2024, *Microbiology Spectrum*) on a smaller spatial scale still reported high sharing of viral taxa, suggesting that it's not

the spatial level that determines the core virome but rather the method on how the virome is looked at (pools vs. individuals). This further shows to us that the core virome concept, according to our definition, does not seem to hold.

Line 68. A quantitative analysis of publications on mosquito viromes showed that the mosquito genus with most publications is the *Culex* genus (Moonen et al, 2023).

Lines 37-38: replaced “*Culex pipiens* virome studies, and virome studies on mosquitoes from the *Culex* genus in general, are underrepresented in publications about the mosquito virome.” with “Virome studies on individual *Culex pipiens*, and on individual mosquitoes in general, have been lacking.”

Line 64: removed ‘virome’

Line 68: replaced “Unfortunately, the *Culex* virome remains understudied” with “Major efforts to sequence the virome of these genera have been made in the past few years.”, and cited Moonen *et al.*

Lines 562-564 "The "core virome" concept was originally coined after a virome study using *Aedes aegypti* and *Culex quinquefasciatus* mosquitoes captured in Guadeloupe,...". In fact, the term "core virome" had already been used in another study (Broecker et al, 2017 Gut Microbes). Moreover, the use of the term "core" was widespread in community ecology long before the paper cited by the authors.

We are aware of the fact that the general term “core virome” was coined before the publication we referred to, but we meant this in the context of the mosquito virome. We changed this in the manuscript to "The "mosquito core virome" concept was originally coined after a virome study using *Aedes aegypti* and *Culex quinquefasciatus* mosquitoes captured in Guadeloupe,...".

Lines 568-569. The authors do not provide a reference and there is a study that obtained results diverging from this sentence (reference 12).

This sentence was posed as a hypothesis to explain our diverging results. Taking reference 12 into account this hypothesis does not seem to hold. We accordingly removed this sentence.

Lines 577-580. There is no need to invoke the Amplification effect hypothesis. Human activity almost always leads to drops in biodiversity, including microorganism diversity. I find difficult to follow this section of the discussion.

Following reviewer #2’s suggestion, this part of the discussion has been removed.

Lines 611-612. The reason behind the limited number of bacteriophage sequences seems not the more plausible and the authors do not provide references supporting it.

We phrased our hypothesis more carefully and provided an extra reference on lines 582-596.

A previous work from the same group found a dominance of bacteriophage sequences in *Culex* mosquitoes (reference 8). Thus, an alternative hypothesis is a sensibility problem.

In light of more recent methods to identify bacteriophage sequences (genomad for identification, CheckV for sequence completeness) the results from the paper to which reviewer 2 refers (Shi *et al.*, 2019), are probably not reliable. The main part of bacteriophage sequences in this paper were very short < 1500bp and are more likely derived from bacteria and/or the bacteriophage genomes were highly fragmented.

After reanalyzing, only 32 out of 299 originally predicted bacteriophage contigs (https://figshare.com/articles/dataset/mosq_phageOTU_fasta/9211592) are predicted by genomad to be viral, and 0 of these genomes are predicted by CheckV to be of good quality (>50% complete). Thus, none of the previously identified bacteriophages are still compliant with the current standards for phage genomes.

We discuss this now on lines 600-604.

- few data on the Culex virome: to nuance: more publications on Culex viruses than the other genus (see Moonen 2023 One Health)

See above.

Re: mSystems00012-24R1 (Lack of abundant core virome in *Culex* mosquitoes from a temperate climate region despite a mosquito species-specific virome)

Dear Prof. Jelle Matthijnssens & Co:

Congratulations! Your manuscript has been accepted by the two reviewers and I, and I am forwarding it to the ASM production staff for publication. Your paper will first be checked to make sure all elements meet the technical requirements. ASM staff will contact you if anything needs to be revised before copyediting and production can begin. Otherwise, you will be notified when your proofs are ready to be viewed.

Cover Image Submissions: If you would like to submit a potential Cover Image, please email a file and a short legend to msystems@asmusa.org. Please note that we can only consider images that (i) the authors created or own and (ii) have not been previously published. By submitting, you agree that the image can be used under the same terms as the published article. Image File requirements: TIF/EPS, 7.5 inches wide by 8.25 inches tall (at least 2,250 pixels wide by 2,475 pixels tall), minimum 300 dpi resolution (600 dpi preferred), RGB, and no figure elements, e.g., arrows or panel labels. The legend should be a short description of the image, 1-2 sentences recommended.

We recognize that the video files can become quite large, so to avoid quality loss ASM suggests sending the video file via <https://www.wetransfer.com/>. When you have a final version of the video and the still ready to share, please send it to mSystems staff at msystems@asmusa.org.

Sincerely,
Seth Bordenstein
Editor

mSystems

Reviewer #1 (Comments for the Author):

The authors have addressed my comments.

Reviewer #2 (Comments for the Author):

I find that the authors have satisfactorily answered most of my remarks. Moreover, I would like to thank the authors for their explanation on what a core virome is, particularly the influenza virus example.